# Interaction of Glia Cells with Glioblastoma and Melanoma Cells under the Influence of Phytocannabinoids

**DOI:** 10.3390/cells11010147

**Published:** 2022-01-03

**Authors:** Urszula Hohmann, Christoph Walsleben, Chalid Ghadban, Frank Kirchhoff, Faramarz Dehghani, Tim Hohmann

**Affiliations:** 1Department of Anatomy and Cell Biology, Medical Faculty, Martin Luther University Halle-Wittenberg, Grosse Steinstrasse 52, 06108 Halle (Saale), Germany; urszula.hohmann@medizin.uni-halle.de (U.H.); cwalsleben@t-online.de (C.W.); chalid.ghadban@medizin.uni-halle.de (C.G.); faramarz.dehghani@medizin.uni-halle.de (F.D.); 2CIPMM, Department of Physiology, Faculty of Medicine, Saarland University, Building 48, 66421 Homburg, Germany; frank.kirchhoff@uks.eu

**Keywords:** THC, CBD, microglia, astrocytes, glioblastoma, melanoma, brain metastasis

## Abstract

Brain tumor heterogeneity and progression are subject to complex interactions between tumor cells and their microenvironment. Glioblastoma and brain metastasis can contain 30–40% of tumor-associated macrophages, microglia, and astrocytes, affecting migration, proliferation, and apoptosis. Here, we analyzed interactions between glial cells and LN229 glioblastoma or A375 melanoma cells in the context of motility and cell–cell interactions in a 3D model. Furthermore, the effects of phytocannabinoids, cannabidiol (CBD), tetrahydrocannabidiol (THC), or their co-application were analyzed. Co-culture of tumor cells with glial cells had little effect on 3D spheroid formation, while treatment with cannabinoids led to significantly larger spheroids. The addition of astrocytes blocked cannabinoid-induced effects. None of the interventions affected cell death. Furthermore, glial cell-conditioned media led to a significant slowdown in collective, but not single-cell migration speed. Taken together, glial cells in glioblastoma and brain metastasis micromilieu impact the tumor spheroid formation, cell spreading, and motility. Since the size of spheroid remained unaffected in glial cell tumor co-cultures, phytocannabinoids increased the size of spheroids without any effects on migration. This aspect might be of relevance since phytocannabinoids are frequently used in tumor therapy for side effects.

## 1. Introduction

Controlling the microenvironment and immune system seem to be promising strategies to treat brain tumors. For both glioblastoma (GBM) and brain metastases, the effect of total neurosurgical resection and other therapy options are limited and survival rates remain very poor. The standard treatment consists of surgery, followed by adjuvant radio- and chemotherapy. Therefore, new effective treatment strategies and targets for GBM and brain metastasis are needed, including therapies targeting the tumor micromilieu. The cellular components of the micromilieu in brain tumors include mainly tumor-associated microglia, macrophages and astrocytes, and cells of the perivascular niche and additional peripheral immune cells [1,2]. The composition of immune cells and activation types within the tumor is dynamic and stage dependent [3].

In GBM, the most malignant form of primary brain tumors, up to 30% of the tumor mass can be comprised of tumor-associated microglia and macrophages (TAM) [4,5]. Interestingly, the amount of TAM negatively correlates with clinical outcome and positively with the staging of glioma [6]. In response to endogenous or exogenous stimuli microglia appear in many distinct stages of activation with different motility, expression of molecules and cytokines. The tumor-associated microglia display an amoeboid morphology similar to activated microglia observed in other pathologies [3]. Untreated primary murine microglia were shown to promote the migration of GL261 mouse glioma cells in Boyden chamber assays. Furthermore, the invasion of GBM cells was significantly reduced in absence of microglia [7,8]. Most effects of microglia on tumor cell migration and motility were associated with secreted soluble factors, such as transforming growth factor-beta (TGF)-β, epidermal growth factor receptor (EGFR) ligands, diverse interleukins, and matrix metalloproteinases (MMP) [9,10,11,12,13]. In addition, untreated microglia reduced sphere formation of brain tumor-initiating cells in culture [14]. Despite microglia, also other cells of glial origin, including astrocytes, interact with and are found within brain tumors. Immunohistochemistry revealed that reactive astrocytes surrounded and infiltrated glioma in human biopsies and murine samples [15,16,17]. Astrocytes had not only glioma-protective effects against chemotherapy but also enhanced tumor invasion into the brain [16,18,19,20,21,22,23]. Furthermore, reactive astrocytes were described as a key component of a tumor-supportive post-surgery microenvironment [24]. Notably, astrocytes were activated by brain metastasis, such as melanoma, facilitating tumor cell invasion into the brain [25,26]. Astrocytes appeared to mediate their effects on the one hand via secreted factors such as TGF-β, interleukins, growth factors, etc. [21,27,28,29,30,31,32], and, on the other hand, via direct connections to tumor cells via gap junctions [20,33,34].

In past studies, cannabinoids that target the cannabinoid receptors CB_1_ and/or CB_2_ were shown to exert anti-tumoral effects in both GBM and melanoma. The two phytocannabinoids tetrahydrocannabinol (THC) and cannabidiol (CBD) reduced the tumor size in glioma xenograft models when applied alone or combined [35,36,37,38,39,40,41,42,43]. Similarly, in melanoma xenografts THC reduced tumor size and proliferation in a CB-dependent manner [44,45]. Despite that, cannabinoid receptor targeting could be associated with altered motility and cell elasticity in GBM cells and others [46,47], thus potentially affecting collective migration as well. Interestingly, both melanoma [48] and GBM [49] possess CB_1_ and CB_2_, while astrocytes express mostly CB_1_ and microglia mainly CB_2_ upon activation [50]. Paired with the anti-tumoral effects of cannabinoids these observations make CBs a potential substance class for targeting tumor cells and interfering with the tumor-stroma-cell cross talk. Currently, the question is not addressed if and how cannabinoids affect tumor-stroma cell cross-talk in the central nervous system.

In this study, the effects of the two phytocannabinoids, namely THC and CBD were evaluated on melanoma or GBM spheroid formation, as well as collective and single-cell migration. Furthermore, the impact of astrocytes or microglia on spheroid formation was analyzed, and the impact of cannabinoids on interactions between tumor and astrocytes or tumor and microglia was investigated.

## 2. Materials and Methods

All experiments involving animal material were performed in accordance with the directive 2010/63/EU of the European Parliament and the Council of the European Union (22 September 2010) and approved by local authorities of the State of Saxony-Anhalt (permission number: I11M18) protecting animals and regulating tissue collection used for scientific purposes.

### 2.1. Cell Culture

Primary microglia and astrocytes were isolated and cultured from C57BL/6J and CX3CR1^GFP/wt^ [51,52] mice as described before [53,54]. After 10–14 days microglial cells were isolated from astrocytic monolayer and used for further experiments. A375 [55] (gifted by Simon Jasinski-Bergner, University Halle-Wittenberg, Halle (Saale), Germany), BV2 microglia [56] (obtained from Ullrich, University of Zürich, Zürich, Switzerland) and primary glial cells were cultured in DMEM (Invitrogen, Carlsbad, CA, USA) supplemented with 10% FBS (Invitrogen) and 1% penicillin/streptomycin (Invitrogen). LN229 [57] were cultured in RPMI medium (Lonza, Basel, Switzerland) with 10% FBS and 1% penicillin/streptomycin. After two days, the medium was collected from confluent astrocytes or BV2 microglia, filtered (Sarstedt, Nümbrecht, Germany), and applied on the tumor cells in a 1:1 ratio with the respective culture medium. This medium was added 3 h before starting the imaging for both single cell and collective migration experiments.

For cannabinoid treatment cannabidiol (5 µM, Tocris Bioscience, Bristol, UK) [39,58,59], tetrahydrocannabinol (5 µM, Tocris) [42] or a combination of both was applied 3 h before the start of the experiments. THC and CBD were both dissolved in DMSO.

### 2.2. Single Cell Migration

For time-lapse microscopy, 4000 cells were seeded in a 12-well plate (Greiner, Kremsmünster, Austria) 24 h prior to the start of the experiments. On the day of experiment, cells were treated with cannabinoids and/or BV2 (BV2CM) or astrocyte conditioned (ACM) media and 3 h later the measurements were performed. Images were obtained every 10 min with a microscope (Leica DMi8, Leica, Wetzlar, Germany) equipped with CO_2_ (5% *v*/*v*) and temperature (37 °C) regulation. The experiments were conducted as described previously [46] and the mean speed of each cell was calculated. Briefly, using the Sobel operator and morphological opening and closing cells were segmented and tracked over time.

### 2.3. Collective Migration

Cells (250,000 A375 or 400,000 LN229) were placed in a 12-well plate to obtain a dense monolayer. On the next day, the treatment with THC and/or CBD and/or BV2 or astrocyte conditioned medium was performed. Three hours later, measurements were started capturing a single image every 3 min.

Velocity fields were calculated using particle image velocimetry [60,61,62], with a cross-correlation window size of 32 × 32 pixels (pixel size: 0.48 µm).

The 4-point susceptibility *χ* was calculated for quantifying size and lifetime of collectively moving cells:$$\chi =N\left[\langle Q{\left(\Delta t\right)}^{2}\rangle -\langle Q{(\Delta t)\rangle }^{2}\right]$$

The peak of *χ* is proportional to the number of collectively moving cells in a dense layer, and the peak position corresponds to the pack life time [63,64]. *Q* is the self-overlap or order parameter given as:$$Q\left(\Delta t\right)=\frac{1}{N}\overset{N}{\underset{i=1}{\sum }}w_{i}\;with\;w=\{\begin{array}{c} 1;if\;\Delta r>0.2d\\ 0;else\; \end{array}$$
where *N* is the cell number and Δ*r* is the cell’s distance to its initial position and d is its diameter. *Q* gives the relative number of cells that moved away more than 20% of their cell size from their initial position.

### 2.4. 3D Spheroid Co-Culture Assay

Three-dimensional (3D) tumor aggregates were generated by using the liquid-overlay method as described before [65]. Cells (20,000) were placed in 96-wells coated with 4% (*w*/*v*) agarose (Peqlab, Erlangen, Germany) and imaged for 72 h, every 15 min Image analysis was performed with self-developed software as described before [65]. Notably, imaging of spheroids started 6 h after cell seeding, because cells were needed to settle down and form an initially loose cell cluster corresponding to the final position of the aggregate inside the well.

Co-culture spheroids were generated with 5%, 10%, 15%, and 30% of astrocytes or CX3CR1^∆/wt^ microglia and tumor cells adding up to a total of 20,000 cells, and the optimal proportion of glia cells was determined. It must be noted that glia and tumor cells were added as a suspension together at the same time to the well plate for the formation of the spheroid. For further experiments, 30% of astrocytes were used since astrocytes made up to 30% of the tumor mass and this percentage showed the largest effects. For microglia, a 10% ratio was chosen since it had the strongest effects, without fully disturbing spheroid formation.

For image analysis of spheroids, a custom-written software written in MatLab (The MathWorks, Natick, MA, USA) was used as described earlier [65]. Briefly, spheroids were segmented using a level set function for tracking each spheroid over time. To determine the size of the spheroid, the amount of its comprising pixels was assessed. After analysis, the spheroid size was normalized to the initial size (0 h) of the control of the respective cell line. For further analysis of growth characteristics, an exponential regime in the area oversize plot was identified manually in the log-log plot and fitted using the following equation:A=A0×e−tt0+A1
where *A* denotes the projected area of the spheroid and *t* the time. The ratio *A*_0_/*t*_0_ can be understood as a characteristic shrinkage rate.

### 2.5. Analysis of Proliferation in 3D Spheroids

Spheroids were removed from culture medium 72 h after seeding on agarose and fixed with 4% PFA. For staining, spheroids were incubated after fixation with succhrose (Carl Roth, Karlsruhe, Germany) and cut on a cryostat (Leica, Wetzlar, Germany) in 12 µm thick slices.

First, normal goat serum was applied for 30 min before incubation with primary antibody overnight (anti-Ki67 antibody for proliferation assessment, DSC innovative Diagnostic System, Hamburg, German). On the next day, washing steps were performed, before application of secondary antibodies (anti-rabbit Alexa Fluor 488, Thermo Fisher Scientific, Waltham, MA, USA) followed by incubation with DAPI (Sigma-Aldrich). Finally, sections were washed and covered with DAKO mounting medium.

After labeling, slices were imaged with a laser scanning microscope (Leica DMi8), using a 63× objective. For proliferation analysis, DAPI and Ki67 images were first denoised using the BM3D filter [66] and subsequently thresholded to obtain binary images of both the DAPI and Ki67 channel. A proliferative index was calculated as the ratio of the number of Ki67 positive pixels that were also DAPI positive relative to the number of DAPI positive pixels.

### 2.6. Flow Cytometry Measurements and Gating Strategies

Spheroids from at least three independent experiments were dissociated with trypsin/EDTA (Thermo Fisher Scientific, Waltham, MA, USA) and propidium iodide (1µL/mL, Miltenyi Biotec GmbH, Bergisch Gladbach, Germany) was added to identify dead cells. All samples were measured (30–80,000 events per panel) by using the flow cytometry analyzer MACS Quant 10 (Miltenyi Biotec GmbH).

The gating strategy is depicted in Appendix A. Data are presented as the percentage of dead cells relative to the overall number of cells measured. Co-expression analyses of GFP and PI were performed in one flow cytometric multi-color panel to prove the presence of dead microglia.

### 2.7. Statistics

Data are presented as the mean standard error of the mean (SEM) of at least three independent experiments. The SEM is depicted either as error bars or shaded areas. Precise sample sizes are given in the Appendix A. Data were analyzed with a one-way ANOVA test with Tukey post test. Differences were considered significant at *p* ≤ 0.05.

## 3. Results

### 3.1. Cannabinoid Induced Slowdown of Spheroid Formation Is Abrogated by Glia Cells

For co-culture experiments evaluating the effect of astrocytes and microglia on spheroid formation, an optimal glial to tumor cell ratio was established with microglia or astrocyte proportions from 5 to 30%. The strongest deviations from control conditions were observed for 10% microglia and 30% astrocytes (Appendix A), and these concentrations were used for further experiments. Notably, the sample size (Appendix A) for these initial tests was lower and after the addition of 30% microglia to A375 cells the spheroid formation was significantly distorted.

Interestingly, both CBD and THC alone inhibited spheroid aggregation in terms of spheroid size in pure A375 melanoma and LN229 GBM cell cultures (Figure 1a–c and Figure 2a–c).

Adding CX3CR1^GFP/wt^ microglia [51] cells alone led to an initial inhibition of the aggregation process, but did not influence the state of equilibrium for spheroid aggregation after 70 h for both tumor cell types (Figure 1e–g and Figure 2e–g). For A375 cells, the addition of both cannabinoids to the microglia-A375 co-culture did not affect initial aggregation speed and only led to larger spheroids in equilibrium after 70 h for all cannabinoid combinations when compared to the microglia-A375 co-cultures (Figure 1e–g). For LN229 cells, adding THC, CBD, or a combination of both to the LN229-microglia co-cultures led to an initial acceleration of spheroid aggregation that could not be observed anymore in equilibrium. CBD or THC alone did not cause an initially increased aggregation and led to an inhibition of aggregation in equilibrium after 70 h compared to the LN229-microglia co-cultures (Figure 1e–g).

Creating astrocyte-tumor co-cultures had no significant effect on spheroid formation in A375 cells but led to an initial acceleration of the aggregation process in LN229 that was not found after 70 h anymore. The addition of astrocytes to both tumor cell types inhibited most of the cannabinoid-induced effects. Significant differences were found in the initial aggregation behavior for astrocyte-A375 co-cultures when treated with CBD and in the equilibrium conditions for astrocyte-LN229 co-cultures treated with CBD, when compared to the untreated co-cultures (Figure 1i–k and Figure 2i–k).

For further analysis of the aggregation dynamics beyond the starting and end point, the aggregation speed was investigated in an exponential shrinkage phase (Appendix A), corresponding to the interval of 2–20 h for LN229 and 15–45 h for A375 cells. Notably, most of the shrinkage of the spheroids occurred in the mentioned time intervals. In LN229 spheroids, both cannabinoids induced faster aggregation when applied alone or in combination, while the addition of astrocytes slowed down spheroid aggregation and blocked the cannabinoid effects in the exponential phase. Furthermore, the addition of microglia cells reduced the spheroid aggregation speed of LN229 cells, but to a lesser extent than astrocytes. Microglia cells also abolished cannabinoid-associated effects on aggregation speed for THC, CBD, and the combined treatment. For the THC+CBD treatment aggregation was—in contrast to the control conditions—slowed down.

For A375, the characteristic aggregation speeds were not influenced by cannabinoids, astrocytes, or cannabinoids + astrocytes. Only microglia slowed down the aggregation speed in A375 cells and this effect was not affected by cannabinoid treatment (Appendix A).

As cannabinoids are known to induce cell death, it was evaluated if any of the chosen treatments caused changes in cell viability, as this may alter spheroid formation. For both cell types and all treatments, we did not find significant changes in cell viability, and cell death rates were between 10–20% for A375 cells and 4–12% for LN229 cells, depending on the exact treatment (Figure 1d,h,k and Figure 2d,h,k). Similarly, cannabinoid treatment is associated with changes in proliferation. Thus, the relative number of cells not being in the G0 phase using Ki67 labeling in spheroid sections was analyzed. The analysis revealed that the overall amount of proliferative cells was very low (<5%), independent of the chosen treatment, and no statistically significant differences were found (Appendix A). Consequently, proliferation is considered to be—at most—of minor importance in this specific model.

As spheroid aggregation, especially in the initial phase, largely depends on the formation of cell–cell contacts and tension, it was next evaluated if cannabinoids and/or supernatants of astrocytes or microglia affect single cell or collective migration.

### 3.2. BV2 and Astrocyte Supernatants Inhibit Collective But Not Single Cell Migration

To assess single tumor cell motility, the application of both CBD and/or THC alone or in combination with BV2 and astrocyte supernatants was performed, corresponding to the same groups as for the spheroid formation assay. Thereby, no statistically significant difference for A375 melanoma and LN229 GBM cells was found (Figure 3a,d,g and Figure 4a,d,g). Next, the collective migration speed of both cell types was analyzed in a dense monolayer under the same conditions. Again, cannabinoids alone did not influence the mean layer migration speed (Figure 3b,c and Figure 4b,c), but the supernatants of both BV2 cells and astrocytes reduced layer speed to 55–65% of the control levels for both tumor types. Notably, this effect was not significantly altered after the addition of both cannabinoids (Figure 3e,f,h,i and Figure 4e,f,h,i).

To further evaluate the origin of changes in migratory behavior, the order parameter and the four-point susceptibility were evaluated. Again no effects of cannabinoids were seen on the order parameter for both cell lines (Figure 5a,c). The application of supernatant resulted in a delayed drop of the order parameter for both cell types (Figure 5e,g,i,k). Such behavior implies that cells stay significantly longer near their initial location and the monolayer shows less reorganization, agreeing with the reduced layer migration speed. From the order parameter, the four-point susceptibility was calculated to analyze the time cells move together as packs (position of the peak) and how many cells move as a pack together (Figure 5b,d,f,h,j,l). After determination of the peak heights, A375 cells were found to move in packs of 3.3 ± 3 cells and all treatments resulted in pack sizes from 2.7 to 6.1 cells, with similar standard deviations. For LN229, the pack size was around 4.1 ± 3 cells and all treatments altered pack size to values from 2.4 to 7 cells, with similar standard deviations. Thus, none of the interventions had a significant impact on the number of collectively moving cells for both tumor entities. Analyzing the peak position cannabinoids showed no significant effect on the time cells move together in a pack for both A375 cells (from 135 min to 123 to 165 min) and LN229 (from 156 min to 147 to 170 min). Supernatants of astrocytes and BV2 cells increased the time A375 cells move together as a pack from 135 min to 225 min or 183 min, respectively. For LN229 cells, a qualitatively similar behavior was found, with both supernatants increased pack life time from 156 min to 291 min or 255 min, respectively (Figure 5b,d,f,h,j,l). Consequently, this analysis supports the results obtained from the layer migration speed and order parameter and shows that A375 and LN229 cells treated with supernatants display statistically less layer reorganization and thus migration. Cannabinoids did not significantly change the effects of supernatants on both tumor cell types. Thus, this data indicates, that both cannabinoids do not influence single-cell or collective migration, while supernatants of BV2 microglia or primary astrocytes inhibit collective but not single-cell migration.

## 4. Discussion

In this study, the functional role of tumor–stroma cell interactions was examined, and the influence of CBD and THC on these interactions was explored. The tumor stroma, the non-neoplastic part of the tumor microenvironment is composed of extracellular matrix and non-neoplastic cells [67]. Here, we focused on interactions between tumor cells and glial cells, especially astrocytes and microglia cells. For both astrocytes and microglia, it is known that they have complementary functions in homeostasis [68]. Furthermore, the glial cells without contact with tumor cells behave differently and have anti-tumorous properties in contrast to tumor-associated cells [69]. Here, microglia and astrocytes affected the initial formation dynamics of 3D aggregates but not the equilibrium conditions (70 h), while cannabinoids tended to hamper aggregate formation. Furthermore, astrocyte and microglia supernatants inhibited collective but not single-cell migration. In addition, cannabinoids did not affect cell migration.

### 4.1. Astrocytes and Microglia Inhibit Initial Spheroid Formation

Both astrocytes and microglia reduced spheroid formation speed in LN229 cells, whereas in A375 co-cultures microglia but not astrocytes reduced the speed of spheroid formation. All treatments did not affect spheroid size at 70 h. During the formation of 3D aggregates, the size and shape of spheroids have been shown to be primarily determined by cell–cell and cell–matrix adhesion and generated tension [70,71,72]. Consequently, the addition of astrocytes or microglia in the current study likely affects the formation of adhesion sites and/or the buildup of tension. As a significant part of the tumor spheroid in co-cultures consisted of stroma cells, it cannot be excluded that the here observed effects are—at least partly—mediated by a physical blockade or different affinities of tumor cell–tumor cell and tumor cell–glial cell adhesions. Following this argument, a linear or bimodal relation between the stroma-to-tumor cell ratio and the spheroid size was expected. Yet, no dependence of aggregate size on glial cell number was observed here Thus, the hindered spheroid aggregation was likely not only caused by physical/space constraints. A further potential explanation might be altered signaling pathways causing changes in tension and/or adhesion. TAM and tumor-associated astrocytes were demonstrated to secrete increased amounts of growth factors, such as EGFR ligands, TGF-β, interleukins, and others [9,10,11,12,13,21,27,28,29,30,31,32]. These molecules were previously demonstrated to induce changes in cytoskeletal organization and expression of adhesion molecules, such as N-cadherin, actin, or vimentin [73,74,75,76,77,78,79]. Despite secreted molecules, direct contacts via gap junctions were found to be a major contributor to astrocyte–tumor interactions. Noteworthy is connexin 43, which interacts with F-actin, β-tubulin, N-cadherin, myosin II, and the actin-binding proteins drebrin und cortactin [80,81,82,83]. Via such signaling, it seems plausible that astrocytes and microglia affected adhesion and tension and thus spheroid formation in this study. Yet, the exact mechanisms need to be elucidated. Our data additionally suggest that the astrocyte/microglia-induced changes affect mostly the adhesion/tension dynamics of spheroid formation, as the effects were only present at the beginning of the measurement and did not persist in equilibrium. Notably, the effects of astrocytes and microglia on spheroid formation and speed were very similar for both, pointing to a more general mechanism although the signaling might be strongly different between melanoma and GBM cells [84,85]. In spheroid-based models, multiple cell types can be cultured together to generate multicellular heterotypic spheroids, accurately recapitulating tumor features including cellular heterogeneity, molecular mechanism, cell–cell/cell–matrix interactions, similar to those under in vivo conditions [67]. In this study, tumor cells and one specific glia cell type were co-cultured to analyze the impact of glia cells on interactions between tumor cells and glia. The model used herein qualifies as an important step in understanding tumor–stroma interactions. Indeed, in more complex organotypic models and tumor slice cultures, additional interactions will potentially yield a more complex behavior as a sum of effects will be detected caused by differential extracellular matrix composition and further cell types, such as e.g., infiltrating immune cells, endothelial cells, etc. However, this complexity will prevent the investigation of the impact of a single cell type.

### 4.2. THC and CBD Inhibit Spheroid Aggregation

The effect of cannabinoids on the final aggregates of melanoma and GBM cells was qualitatively the same, as the treatment with CBD and THC led to larger aggregates. In terms of dynamics cannabinoids induced faster aggregation in LN229 cells, but not in A375, while the addition of glial cells tended to reduce the aggregation speeds in both cell lines. GBM and melanoma cells express both cannabinoid receptors [46,49,86]. Both cannabinoids appeared to change the ratio of tension to adhesion in favor of tension. A previous study demonstrated changes in cell-elastic modulus and reduction in adhesion in GBM cells after targeting CB_1_ or CB_2_ [46]. Both effects might explain the observed larger spheroid sizes and altered dynamics. Furthermore, cannabinoids—including THC—were found to affect signaling cascades relevant for cytoskeletal organization and adhesion formation, such as decreased FAK phosphorylation in mammary carcinoma cells [87,88] or increased FAK phosphorylation in lung carcinoma cells [89]. Moreover, in PC12, Chinese hamster ovarian and neuroblastoma cells THC and CBD reduced the levels of β-tubulin and β-actin and/or induced changes in cytoskeletal organization [90,91,92]. Additionally, the CB_1_/CB_2_ agonist HU210 induced significant reorganization of the actin and microtubule cytoskeleton and reduced expression of β-tubulin and β-actin in PC12 cells [93]. These observations are in agreement with our previous results demonstrating cell-type and receptor-dependent changes in actin cytoskeleton organization after targeting CB_1_ or CB_2_ [94]. All mentioned signaling routes may affect tension and adhesion formation and thus be potential candidates explaining the observed effects of cannabinoids on spheroid formation.

Another issue causing an increase in spheroid size may be increased proliferation. Nevertheless, past studies reported anti-proliferative effects of both THC and CBD, often for significantly higher doses [48,95,96]. In our model system, we did not see significant proliferative activity in spheroids with on average less than 5% of cells not being in the G0 phase, independent of the used treatment. Consequently, effects associated with proliferation or apoptosis appear unlikely as cause for changes in spheroid aggregation.

The effects of phytocannabinoids are dependent on their preparation, concentration, the treated cell type, and the abundance of receptor targets. In GBM, both THC and CBD have been shown to activate in part similar targets and include several cellular pathways which are possibly involved in the regulation of spheroid size [95] as observed here. CBD modulated the activation of pAKT, mTOR, pERK, β-catenin, PLCG1, and p38 MAPK, and pSTAT3 [43,97]. THC, CBD, or a combination of both reduced the activation of pAKT [98]. Administration of THC inhibited MMP-2 expression in an in vivo model of glioma [35,99,100]. Furthermore, THC induced phosphorylation of eukaryotic translation initiation factor 2α (eIF2α) followed by activation of an ER stress response that promoted autophagy via tribbles homolog 3-dependent (TRB3-dependent) and inhibited Akt/mammalian target of rapamycin complex 1 (mTORC1) axis [42].

Little is known about the effects of THC or CBD on signaling cascades in melanoma cells. Akt was involved in the inhibition of melanoma cell proliferation after THC treatment, while ERK, JNK, and p38 MAPK were not significantly affected [48]. THC-induced autophagy was not prevented by knockdown of Beclin-1, suggesting that in contrast to glioma, noncanonical autophagy-mediated apoptosis in response to THC in melanoma [42,45,100]. In the current study, a highly complex system was used containing GBM or melanoma cells in co-culture with glial cells treated with up to two cannabinoids. This complex model makes the comparison with other in vitro models very difficult and needs a systematic analysis of intracellular pathways in order to better understand the tumor micromilieu and effects of cannabinoids.

Interestingly, when cannabinoids were applied to tumor-astrocyte co-cultures most cannabinoid-associated effects were abolished, while for tumor-microglia co-cultures most effects persisted. Recent studies demonstrated the formation of gap junctions between astrocytes and melanoma and GBM cells mediating chemo-protective effects in a potentially calcium-dependent manner [32,101]. Moreover, astrocytes were demonstrated to actively rescue GBM cells from apoptosis [102]. Comparable mechanisms might be responsible for the absence of effects in the astrocyte co-cultures treated with cannabinoids. Opposing effects of TAM have been reported in different tumor types. They have been shown to be partly responsible for resistance to classical anti-tumor treatments, but also to improve treatment efficacy [103,104].

### 4.3. Supernatants of Astrocytes and Microglia But Not THC or CBD Inhibit Collective Migration

As the results from spheroid aggregation experiments hint towards changes in adhesion and/or tension dynamics we evaluated the effects of astrocyte or microglia supernatants on single cell and collective migration. Interestingly, for both cell types, supernatants of microglia and astrocytes did not influence single-cell motility but reduced the collective migration speed, implying effects on cell–cell interactions. Furthermore, supernatants triggered both melanoma and GBM cells to move a prolonged time together, indicating that the type of motion inside the layer became slower but more persistent. Responsible molecular processes might be similar to those discussed for the spheroid aggregation experiments, involving growth factors, such as EGFR-ligands, TGF-β, interleukins, and others [9,10,11,12,13,21,27,28,29,30,31,32]. Interestingly, the reduction in collective migration after incubation with microglia or astrocytes supernatant seems to contradict previously published results, demonstrating that both astrocytes and microglia favor tumor migration and infiltration [7,8,24]. These differences might arise from the differences in the used models. In studies citing the Boyden chamber, scratch assays or brain slice cultures were used. While Boyden chambers are mainly chemotaxis driven, scratch assays are strongly affected by proliferation and (single) cell migration. The model used here does not contain large-scale spatial or chemical inhomogeneity, and thus is not affected by chemotaxis and less impacted by proliferation. The last part is noteworthy, as both astrocytes and microglia increased the proliferation of tumor cells [101,105]. Contrary to these effects, we did not observe any effects of THC or CBD on collective cellular motion independent of the presence of astrocyte or microglia supernatants. Thus, from a functional perspective, these cannabinoids only appear to affect tumor cohesiveness.

## 5. Conclusions

In this study, we demonstrated astrocytes and microglia cells to slow down the initial 3D aggregate formation of melanoma and GBM cells, as well as inhibiting collective cellular migration speed. Yet, supernatants of astrocytes and microglia led to a more directed collective motion with cells moving for a prolonged time together. Furthermore, THC and CBD were shown to slow down the spheroid formation of melanoma or GBM cells but these effects were absent when astrocytes were co-cultured. THC and CBD did not affect collective migration of both cell types. Thus, our results imply on the one hand that astrocyte or microglia secreted factors impact tumor cell migration. On the other hand, astrocytes seem to hamper the effects of cannabinoids.

Taken together, the current study provides an important and necessary basis for further molecular analysis of the interactions of glioblastoma/melanoma cells and the brain micromilieu, as well as the influence of cannabinoids in this system. Described effects should be evaluated in further model systems, such as organoids, patient-derived cells, and slices. Furthermore, the presented results largely rule out signaling cascades associated with proliferation or cell death in here investigated models, as no effects on these parameters were observed.

## Figures and Tables

**Figure 1 cells-11-00147-f001:**
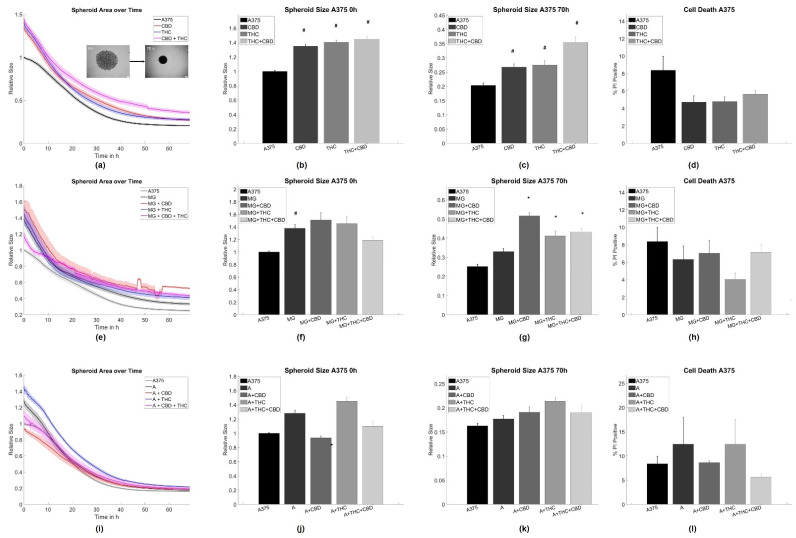
Aggregation results for A375 melanoma cells. (**a**–**c**) Average time evolutions of spheroid sizes for A375 cells treated with cannabinoids. The inlet depicts a typical spheroid aggregation. (**d**,**h**,**l**) Ratio of cell death in A375 spheroids treated with cannabinoids. (**e**–**g**) Time evolution of tumor-microglia (MG) spheroid sizes and cell death for co-cultures with and without cannabinoid treatment. (**i**–**k**) Time evolution of tumor-astrocyte (A) spheroid sizes and cell death for treatment co-cultures with and without cannabinoid treatment. Relative size: Values are normalized to the respective untreated control cell line at time point 0 h referring to the start of the measurement. Stars (*) depict significant results against the spheroids mixed with astrocytes or microglia, respectively. Hashes (#) depict significant results against the untreated control. Error bars and shaded areas depict the standard error of the mean. Abbreviations: CTL: control, CBD: cannabidiol, THC: tetrahydrocannabinol, A: astrocytes, MG: microglia.

**Figure 2 cells-11-00147-f002:**
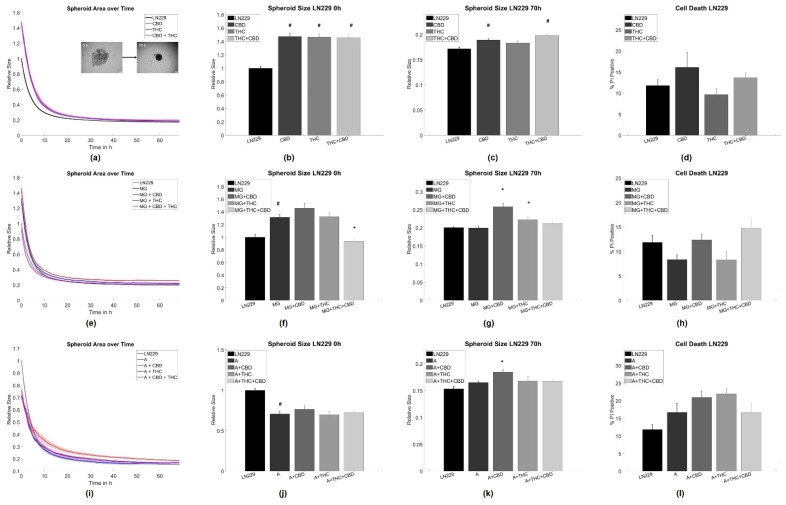
Aggregation results for LN229 GBM cells. (**a**–**c**) Average time evolutions of spheroid sizes for LN229 cells treated with cannabinoids. The inlet depicts a typical spheroid aggregation. (**d**,**h**,**l**) Ratio of cell death in LN229 spheroids treated with cannabinoids. (**e**–**g**) Time evolution of tumor-microglia (MG) spheroid sizes and cell death for co-cultures with and without cannabinoid treatment. (**i**–**k**) Time evolution of tumor-astrocytes (A) spheroid sizes and cell death for co-cultures with and without cannabinoid treatment. Relative size: Values are normalized to the respective untreated control cell line at time point 0 h referring to the start of the measurement. Stars (*) depict significant results against the spheroids mixed with astrocytes or microglia, respectively. Hashes (#) depict significant results against the untreated control. Error bars and shaded areas depict the standard error of the mean. Abbreviations: CTL: control, CBD: cannabidiol, THC: tetrahydrocannabinol, A: astrocytes, MG: microglia.

**Figure 3 cells-11-00147-f003:**
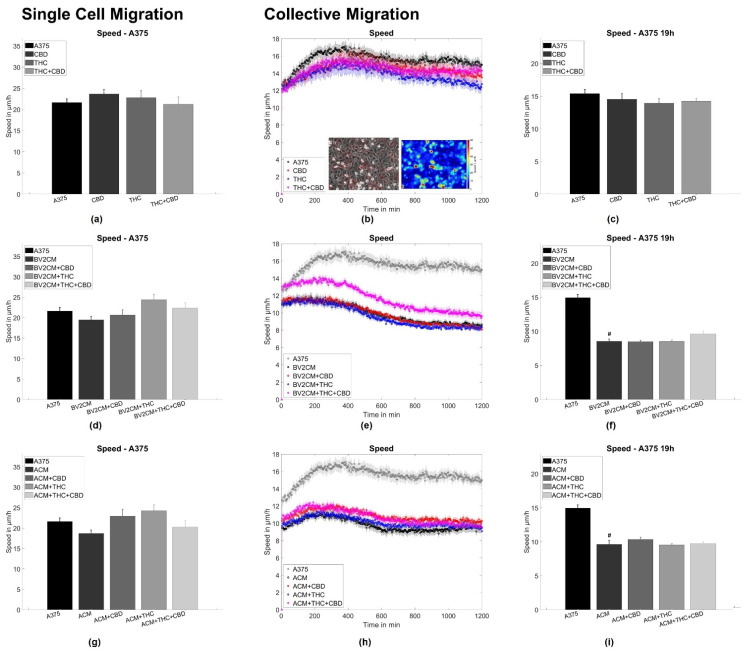
Single cell and collective motion of A375 melanoma cells. (**a**) Single-cell motility of A375 cells treated with cannabinoids. (**b**) Collective migration speed for A375 cells treated with cannabinoids. The inlet shows a typical phase contrast image, overlaid with the associated velocity vectors and a heat map of the local speed overlaid with the velocity vectors. (**c**) Shows the averaged collective migration speeds from 18–20 h of A375 cells treated with cannabinoids. (**d**–**f**) Same measurements as shown in (**a**–**c**) but for A375 cells treated with microglia conditioned media and cannabinoids. (**g**–**i**) Same measurements as shown in (**a**–**c**) but for A375 cells treated with microglia conditioned media and cannabinoids. Hashes (#) depict significant results against the untreated control. Error bars and shaded areas depict the standard error of the mean. Abbreviations: CTL: control, CBD: cannabidiol, THC: tetrahydrocannabinol, ACM: astrocyte conditioned media, BV2CM: BV2 cell-conditioned media.

**Figure 4 cells-11-00147-f004:**
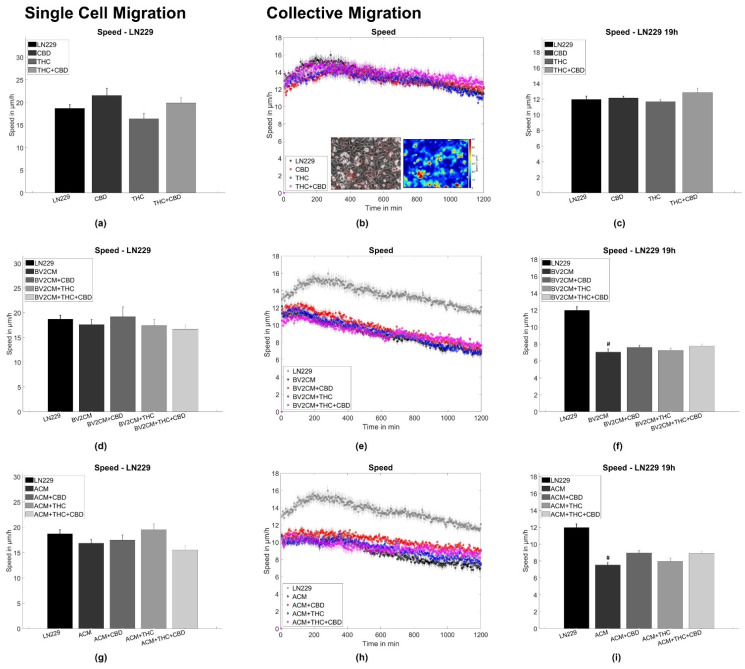
Single cell and collective motion of LN229 GBM cells. (**a**) Single-cell motility of LN229 cells treated with cannabinoids. (**b**) Collective migration speed for LN229 cells treated with cannabinoids. The inlet shows a typical phase contrast image, overlaid with the associated velocity vectors and a heat map of the local speed overlaid with the velocity vectors. (**c**) Shows the averaged collective migration speeds from 18–20 h of LN229 cells treated with cannabinoids. (**d**–**f**) Same measurements as shown in (**a**–**c**) but for LN229 cells treated with microglia conditioned media and cannabinoids. (**g**–**i**) Same measurements as shown in (**a**–**c**) but for LN229 cells treated with microglia conditioned media and cannabinoids.. Hashes (#) depict significant results against the untreated control. Error bars and shaded areas depict the standard error of the mean. Abbreviations: CTL: control, CBD: cannabidiol, THC: tetrahydrocannabinol, ACM: astrocyte conditioned media, BV2CM: BV2 cell-conditioned media.

**Figure 5 cells-11-00147-f005:**
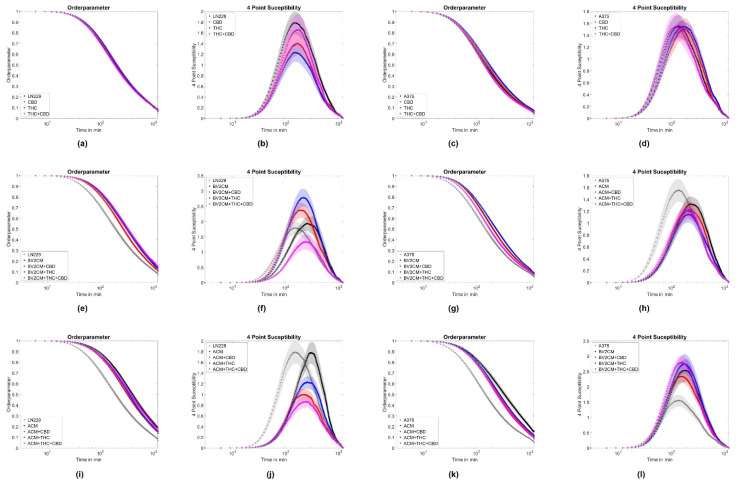
Collective motion parameters for LN229 and A375 cells. (**a**,**e**,**i**) represent the average order parameter *Q* for LN229 cells, when treated with cannabinoids, astrocyte supernatants and cannabinoids or microglia supernatants and cannabinoids. (**b**,**f**,**j**) show the associated four-point susceptibility for LN229 cells. (**c**,**g**,**k**) represent the average order parameter *Q* for A375 cells, when treated with cannabinoids, astrocyte supernatants, and cannabinoids or microglia supernatants and cannabinoids. (**d**,**h**,**l**) show the associated four-point susceptibility for A375 cells. Shaded areas depict the standard error of the mean. Abbreviations: CTL: control, CBD: cannabidiol, THC: tetrahydrocannabinol, ACM: astrocyte conditioned media, BV2CM: BV2 cell-conditioned media.

## Data Availability

All datasets generated for this study are included in the article/Appendix A.

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
