# Peer review of "Interaction of Glia Cells with Glioblastoma and Melanoma Cells under the Influence of Phytocannabinoids"

_cells, 2022, doi:10.3390/cells11010147_

Round 1

Reviewer 1 Report

In the article entitled “Interaction of glia cells with glioblastoma and melanoma cells under influence of phytocannabinoids”, Hohmann et al. investigated the effects of phytocannabinoids on the interactions between microglia or astrocytes and LN229 glioblastoma or A375 melanoma cells. The analysis were performed in 3D spheroid formation and single or collective cell migration. Overall, I believe the results, although quite interesting, have some flaws that need to be properly corrected. Thus, a major revision is needed. My comments are listed below:

1- Spheroid area and size: How were spheroids measured over time? There was no detailed information of the experiment in materials and methods. Also, the plot y-axis in Figures 1 and 2 represent spheroid area and relative size. Are these similar measurements? Spheroid area and relative sizes seem to be normalized. If so, normalized by what? Normalization was never mentioned. The curves in Figures 1 and 2 are averages of several measurements? What are the shadows of each curve? None of these information were mentioned in the manuscript and make it difficult to understand the results 

2- Also, is there any explanation for the relative sizes of treatments to be different from their controls at 0h? If there is a variation in size at 0h, it is probably not because of the treatments but because of sample variability. It also shows that normalization (or the way it was performed) is not appropriate. 

3- Is it possible to determine spheroid aggregation speed? This data would be extremely important for the manuscript. Eventually, using fittings for the 3D spheroid formation curves, this result could be obtained.

Reviewer 2 Report

I believe the manuscript entitled “Interaction of glia cells with glioblastoma and melanoma cells under influence of phytocannabinoids” is suitable for publication after minor alterations. The authors presented clear results that show that astrocytes and microglia alter 3D spheroid formation and inhibit collective cellular migration speed of melanoma and GBM cells. Moreover, they discovered that the cannabinoids tetrahydrocannabinol and cannabidiol do not interfered on 3D spheroid formation or migration of melanoma/GBM cells. This brings an important discussion for the field of study.

Overall English language and style are fine, but minor corrections are needed. Some parts of the text are a bit confusing to read, due to style of writing.

Moreover, supplementary figure S1 has two graphs with the same title: “Spheroid size A375 70h”.

Finally, I would ask that the authors include the measure unit for the spheroid area in the y-axis, or indicate they are arbitrary units, if that is the case.

Author Response

I believe the manuscript entitled “Interaction of glia cells with glioblastoma and melanoma cells under influence of phytocannabinoids” is suitable for publication after minor alterations. The authors presented clear results that show that astrocytes and microglia alter 3D spheroid formation and inhibit collective cellular migration speed of melanoma and GBM cells. Moreover, they discovered that the cannabinoids tetrahydrocannabinol and cannabidiol do not interfered on 3D spheroid formation or migration of melanoma/GBM cells. This brings an important discussion for the field of study.

Overall English language and style are fine, but minor corrections are needed. Some parts of the text are a bit confusing to read, due to style of writing.

We carefully re-read the manuscript and changed ambiguities.

Moreover, supplementary figure S1 has two graphs with the same title: “Spheroid size A375 70h”.

We thank the reviewer for pointing out this error. The labelling of one graph has been corrected to “Spheroid size A375 0h”.

Finally, I would ask that the authors include the measure unit for the spheroid area in the y-axis, or indicate they are arbitrary units, if that is the case.

The y-axis labeling has been changed to “Relative Size” to indicate that the measure is normalized to the control at time point 0 h and thus is dimensionless. Furthermore, this information was added to the methods section.

Reviewer 3 Report

In the article, entitled “Interaction of glia cells with glioblastoma and melanoma cells

under influence of phytocannabinoids”, the authors Urszula Hohmann and co-workers,

 have aimed to analyse the effects of brain parenchyma on one LN229 glioblastoma (primary) tumour cell line and one melanoma A375 cell line, which would when metastasizing to the brain, encounter such microenvironment. However, the tumour microenvironment is only represented by either astrocytes or microglial cells. They analysed motility, as collective movement of tumour cells in pre-formed aggregates grown on agar. Also, global inter-tumour cell(-cell) interactions were assessed  in terms of measuring the sporadic aggregation size in a 3D “model” that they not very appropriately termed “spheroid”.

First, they demonstrated that astrocytes and microglia cells tend to slow down initial 3D aggregate formation of melanoma and glioblastoma cells, as well as inhibiting collective cellular migration speed.  Furthermore, the effects of phytocannabinoids, cannabidiol (CBD), tetrahydrocannabidiol (THC) or their co-application /addition to the aggregates were analysed but no effects were observed, phytocannabinoids seemed to increase the size of spheroids without any effects on migration. Also, no effects of THC or CBD on collective cellular motion independent on the presence of astrocyte or microglia supernatants. THC and CBD did not affect collective migration of both cell types Thus, from a functional perspective these cannabinoids only appear to affect tumour cohesiveness.

Major comments and drawback for this manuscript not to be published in the present format are;:.

  • The 3 D cell cultures and co-culture model is not related to the in vivo tumour microenvironment (TME), as it was only was mimicked by “adding” 30% of astrocytes and separately 5 % of microglial cells for an arbitrary time period, what the authors called “spheroids” although these were just multicellular aggregates, due to the diffusion f astrocytes /microglia into tumour cell aggregates. There is no effort to image and follow the formations  end heterotypic cellular interactions by specific/selective cell biomarkers, nor  membrane adhesion proteins or gap-junctions. Rather low proliferation marker within a “sectioned spheroid” is rather strange (as shown on Figure S3).  Advanced 3 D tumour models are using: (a)n  the tumour spheroid or tumoroshere , grown from cancer stem cells, differentiating into specific tumour type/ subtype of the same origin as primary tumour; (b) Organotypic tumour tissue slices and c)  organoids from the whole tumours grown under cancer stem cell selective or even not highly -selective  conditions  there are a number excellent recent publication on GBM (see Jacob et al., Cell, 2019)  - are the only appropriate  model for  studying the effects of drugs /  phytocannabinoids in whole tumour -including TME.

The group by Hohmann et al.  published several excellent papers on the effect of cannabinoid agonists/antagonists on GBM /single cells, but the model taken here is inappropriate as explained above,  and cannot reach the aim of this study!

  • Secondly, the concentration of TCH and CBD take here is much too low to have  any  Only the experimental  part, when  supernatants of astrocytes and microglia were added to the aggregates present a more defined cellular system and indeed a more directed collective motion with cells moving for a prolonged time together was noticed.
  • Only the experimental part, when supernatants of astrocytes and microglia wee added to the aggregates presents a more defined cellular system and indeed a more directed collective motion with cells moving for a prolonged time together was noticed. Such  “paracrine” interaction via chemokines /receptors signalling axis may be relevant, as stated by the authors.  These results implied  on the one hand that astrocyte or microglia secreted factors that  impact tumour cell migration. However, no  effort has been  made to analyse the cell media that were added and for sure differ for astrocytes and microglia cultures. This aspect is perhaps worth to follow, providing that correct tumorosphere model would be used.

Minor comment;.

  1. The experimental / methodological part, In particular co-cultures data are is rather superficially described, although the methodology is  cited as. previous work of the authors or literature data, so it was harder for the reader to follow the experimental design.

For example:. On page 2. line 90 : Notably, the sample size (supplementary Table 1) for these initial tests was lower and after addition of 30% microglia to A375 91 cells the spheroid formation was significantly distorted. It  is not possible to understand  the data on Table 1 in  the Supplements: there is no Legend to the Table 1, no abbreviations are explained  the numbers listed have no unites of the measurements /no meaning of the numbers?  

Author Response

In the article, entitled “Interaction of glia cells with glioblastoma and melanoma cells under influence of phytocannabinoids”, the authors Urszula Hohmann and co-workers, have aimed to analyse the effects of brain parenchyma on one LN229 glioblastoma (primary) tumour cell line and one melanoma A375 cell line, which would when metastasizing to the brain, encounter such microenvironment. However, the tumour microenvironment is only represented by either astrocytes or microglial cells. They analysed motility, as collective movement of tumour cells in pre-formed aggregates grown on agar. Also, global inter-tumour cell(-cell) interactions were assessed  in terms of measuring the sporadic aggregation size in a 3D “model” that they not very appropriately termed “spheroid”.

First, they demonstrated that astrocytes and microglia cells tend to slow down initial 3D aggregate formation of melanoma and glioblastoma cells, as well as inhibiting collective cellular migration speed.  Furthermore, the effects of phytocannabinoids, cannabidiol (CBD), tetrahydrocannabidiol (THC) or their co-application /addition to the aggregates were analysed but no effects were observed, phytocannabinoids seemed to increase the size of spheroids without any effects on migration. Also, no effects of THC or CBD on collective cellular motion independent on the presence of astrocyte or microglia supernatants. THC and CBD did not affect collective migration of both cell types Thus, from a functional perspective these cannabinoids only appear to affect tumour cohesiveness.

Major comments and drawback for this manuscript not to be published in the present format are;:.

Before coming to the actual points of criticism we want to respond to the following statement that was made when summarizing the manuscript content, as it does not re-occur in the later points of criticism: “They analyzed motility, as collective movement of tumor cells in pre-formed aggregates grown on agar.”

We have to denote that this description does not describe what has been done for the analysis of migration. For clarification: We analyzed single cell motility using sparsely seeded cells and measured the speed of single cells. For collective migration the migration pattern of a dense monolayer was analyzed using particle image velocimetry, a state of the art method for this purpose. 3d spheroids were seeded on agarose and further analyzed.

  • The 3 D cell cultures and co-culture model is not related to the in vivo tumour microenvironment (TME), as it was only was mimicked by “adding” 30% of astrocytes and separately 5 % of microglial cells for an arbitrary time period, what the authors called “spheroids” although these were just multicellular aggregates, due to the diffusion f astrocytes /microglia into tumour cell aggregates. There is no effort to image and follow the formations  end heterotypic cellular interactions by specific/selective cell biomarkers, nor  membrane adhesion proteins or gap-junctions. Rather low proliferation marker within a “sectioned spheroid” is rather strange (as shown on Figure S3).  Advanced 3 D tumour models are using: (a)n  the tumour spheroid or tumoroshere , grown from cancer stem cells, differentiating into specific tumour type/ subtype of the same origin as primary tumour; (b) Organotypic tumour tissue slices and c)  organoids from the whole tumours grown under cancer stem cell selective or even not highly -selective  conditions  there are a number excellent recent publication on GBM (see Jacob et al., Cell, 2019)  - are the only appropriate  model for  studying the effects of drugs /  phytocannabinoids in whole tumour -including TME.

For answering these specific points of criticism the response has been split into parts dealing with each point individually:

Comment: “The 3 D cell cultures and co-culture model is not related to the in vivo tumour microenvironment (TME), as it was only was mimicked by “adding” 30% of astrocytes and separately 5 % of microglial cells for an arbitrary time period […]”.

Response: While it is obviously true that the microenvironment is more complex in nature than can be depicted by the analysis of the interactions of microglia/astrocytes and tumor cells, this claim has never been made by the authors. Regarding the “arbitrary time period” it becomes clear from the spheroid formation assay that at the end of the measurement spheroids reached equilibrium and did no longer change in size significantly. Consequently, all relevant dynamics associated with spheroid formation was captured during the here investigated time interval. Based on our measurements of the very low proliferation index found in the spheroids, a significant growth of the spheroids is expected for significantly higher measurement times in the order of weeks. Yet, the time frame of weeks is not relevant for other measurements taken (collective and single cell migration), as the observed effects were visible from the very beginning of the experiments and persisted for the whole measurement duration. Therefore, we consider the used time frame of the tumor-glia-cell culture adequate and not arbitrary.

Comment: “[…] what the authors called “spheroids” although these were just multicellular aggregates, due to the diffusion f astrocytes /microglia into tumour cell aggregates.”

Response: We disagree with the reviewer’s statement. In this manuscript we used the definition of spheroids presented in the following papers:

From Han et al., 2021 (https://doi.org/10.1186/s12935-021-01853-8)

“MCTs are cell clusters formed by either self-assembly or forced growth starting from single-cell suspensions. The cells are closely packed with high density in spheroids. Therefore, the cells in MCTs communicate strongly and sustain complex communication between cells and extracellular matrix (ECM) [4].”

From “Biomaterials for 3D Tumor Modelling”, Chapter 9 “Metastasis in three-dimensional biomaterials”, by Kundu et al., 2020 (https://doi.org/10.1016/B978-0-12-818128-7.00009-5)

“Spheroids are defined as cellular aggregates obtained in suspension or embedded within 3D matrix [84].”

From Rodrigues et al., 2018 (https://doi.org/10.1016/j.pharmthera.2017.10.018)

“MCTSs are described as spherically symmetric aggregates of cells analogous to tissues, with no artificial substrate for cell attachment.”

Based on these definitions our models can be considered spheroids.

We also have to clarify how spheroids in co-culture with microglia or astrocytes were generated. A suspended mixture of glia and tumor cells was used for the generation of one single spheroid consisting of both glia and tumor cells from the very beginning. To reduce ambiguities on this end we added the following sentence to the respective part of the method section: “It has to be denoted that glia and tumor cells were added as a suspension together at the same time to the well plate for the formation of the spheroid.”

Comment: “Rather low proliferation marker within a “sectioned spheroid” is rather strange (as shown on Figure S3).”

Response: In the described experiments it was found that ≈1-5% of cells were Ki67 positive and thus capable of division. While this number may appear very low it is well reflected by the aggregation dynamics of the spheroids, that reached in the case of LN229 an equilibrium size after approximately 20 h and spheroid size did not increase the next 50 h afterwards. Similarly, for A375 cells, that reached equilibrium after approximately 50 h and did not show growth signs for the remaining 20 h afterwards. Thus proliferation can be assumed to be rather low from this point of view as well. In line with our results, other groups found low proliferative indices (<10%) in spheroids as well (https://doi.org/10.1038/sj.onc.1205053, https://doi.org/10.1093/jnci/46.1.113). Notably, proliferation index is expected to decline with rising spheroid size, as nutrition and oxygen supply in inner parts of the spheroid declines and no supply via blood vessels exists, as also observed by one of the mentioned studies (https://doi.org/10.1093/jnci/46.1.113). For the experiments presented here the final spheroid sizes were ≈600µm in diameter.

Comment: “Advanced 3 D tumour models are using: (a)n  the tumour spheroid or tumoroshere , grown from cancer stem cells, differentiating into specific tumour type/ subtype of the same origin as primary tumour; (b) Organotypic tumour tissue slices and c)  organoids from the whole tumours grown under cancer stem cell selective or even not highly -selective  conditions  there are a number excellent recent publication on GBM (see Jacob et al., Cell, 2019)  - are the only appropriate  model for  studying the effects of drugs /  phytocannabinoids in whole tumour -including TME.”

Response: We first want to emphasize that this study was not designed to evaluate the whole microenvironment but rather a specific type of interactions, namely tumor cells with astrocytes/microglia and if/how cannabinoids interfere.

We fully agree that the mentioned models are more physiological, but this study aims for a first screening if and how cannabinoids interact with certain parts of the tumor microenvironment in a highly reproducible model system using standardized cell lines together with glial cells in a controlled defined number. This approach allows not only for high reproducibility but also for a significant throughput that cannot easily be achieved using primary tumor material. Please denote, that for this specific study the aggregation process of 1004 spheroids was analyzed in a time resolved manner (≈300.000 images; see supplemental table 1), necessitating models that can be generated in large numbers.

The group by Hohmann et al.  published several excellent papers on the effect of cannabinoid agonists/antagonists on GBM /single cells, but the model taken here is inappropriate as explained above,  and cannot reach the aim of this study!

  • Secondly, the concentration of TCH and CBD take here is much too low to have  any  Only the experimental  part, when  supernatants of astrocytes and microglia were added to the aggregates present a more defined cellular system and indeed a more directed collective motion with cells moving for a prolonged time together was noticed.

Regarding the concentrations of THC and CBD (both 5 µM) used here we have to emphasize several different aspects. For reasons of comparability two approved cannabinoid based medications were used as reference. First sativex, an approximately even mixture of CBD and THC, that is administered as a spray, containing 2.7mg THC and 2.5mg CBD per single spray and is recommended to be used at most 12 times a day (see: https://www.medicines.org.uk/emc/product/602/smpc#gref). Assuming an adult of weight 80kg (approximately corresponding to 80 l volume) this would add up to 1.28µM THC dosage, assuming equal distribution of cannabinoids throughout the body and intake of the whole daily dose at once. Similarly, for Cesamet administered orally, containing the synthetic THC-analogue nabilone a daily dose of up to 4mg is recommended (see: https://www.accessdata.fda.gov/drugsatfda_docs/label/2006/018677s011lbl.pdf) resulting in a total concentration of 0.16µM, using the same assumptions as before. It should be considered that because of the lipophilic nature of compounds, higher – presumably 3-4 times - tissue concentrations will be achieved in the brain. Yet, while these small calculations are just approximations they imply that the dosages used are not too low, because similar or lower doses are used in clinical context. Furthermore, this aspect is highly relevant since phytocannabinoids are frequently used in the tumor therapy for side effects.

Furthermore, in previous studies it was demonstrated that the used concentrations are very well feasible to induce statistically significant effects on glioblastoma cells and others (doi: 10.3390/cancers13051064, 10.1158/0008-5472.CAN-05-4566, 10.1371/journal.pone.0076918, 10.1158/1535-7163.MCT-09-0407, 10.1111/j.1476-5381.2011.01497.x, 10.1096/fj.06-6638fje).  Noteworthy, there are suggested synergetic effects observed for the combination of THC and CBD in GBM cells for even lower doses (doi: 10.1158/1535-7163.MCT-10-0688).

Following these two independent lines of arguments, we are convinced that the concentrations used here are not too low.

  • Only the experimental part, when supernatants of astrocytes and microglia wee added to the aggregates presents a more defined cellular system and indeed a more directed collective motion with cells moving for a prolonged time together was noticed. Such  “paracrine” interaction via chemokines /receptors signalling axis may be relevant, as stated by the authors.  These results implied  on the one hand that astrocyte or microglia secreted factors that  impact tumour cell migration. However, no  effort has been  made to analyse the cell media that were added and for sure differ for astrocytes and microglia cultures. This aspect is perhaps worth to follow, providing that correct tumorosphere model would be used.

We agree with the reviewer that the interactions observed between glia and tumor cells are highly interesting and need further investigation.

The focus on the current manuscript is to elucidate the cellular interactions between tumor and glia cells with special attention to cannabinoid effects on spheroid formation. Supernatant experiments will allow statements on the effects of soluble factors secreted from glial cells but ignore the influence of cell-cell-interactions regulated by membrane bound receptors. We think that the conditions in our model are superior to the sole use of supernatants. Glial cells interact on the one hand directly with tumor cells and on the other secrete their factors in intercellular space. Furthermore, the experimental design is highly standardized and can be repeated and reproduced elsewhere. The model systems suggested by the reviewer are far more interference prone and less stable, even though more physiological. The induction, stability and purity in generation of tumor stem cells is highly condition dependent and interference prone. Furthermore, primary models are highly heterogenous, with undefined amounts of infiltrating cells and tumor cell populations. GBM is characterized by high intra- and intertumor heterogeneity. Thus, organotypic primary tumor slices are less suitable for screening a high number of experimental groups due to the limited availability. Consequently, there is no single perfect model. We have therefore added the limitations of the used method to the discussion.

Minor comment;.

  1. The experimental / methodological part, In particular co-cultures data are is rather superficially described, although the methodology is  cited as. previous work of the authors or literature data, so it was harder for the reader to follow the experimental design.

For example:. On page 2. line 90 : Notably, the sample size (supplementary Table 1) for these initial tests was lower and after addition of 30% microglia to A375 91 cells the spheroid formation was significantly distorted. It  is not possible to understand  the data on Table 1 in  the Supplements: there is no Legend to the Table 1, no abbreviations are explained  the numbers listed have no unites of the measurements /no meaning of the numbers?  

We have improved the readability of the methods section. For the given example we have to denote that the heading of the table is “Sample size”, and thus shows the dimensionless number of samples analyzed for each experimental type and treatment group to come to the respective values/data presented throughout the manuscript. The abbreviations used in figures and tables have been introduced in the figure legends.

Round 2

Reviewer 1 Report

The authors have addressed all the points raised by this reviewer.